# Evaluation of a Proportional Response Addition Approach to Mixture Risk Assessment and Predictive Toxicology Using Data on Four Trihalomethanes from the U.S. EPA’s Multiple-Purpose Design Study

**DOI:** 10.3390/toxics12040240

**Published:** 2024-03-25

**Authors:** Linda K. Teuschler, Richard C. Hertzberg, Anthony McDonald, Yusupha Mahtarr Sey, Jane Ellen Simmons

**Affiliations:** 1LK Teuschler & Associates, St. Petersburg, FL 33707, USA; lindateuschler@gmail.com; 2Biomathematics Consulting, Atlanta, GA 30322, USA; 3Public Health and Integrated Toxicology Division, Center for Public Health and Environmental Assessment, Office of Research and Development, U.S. Environmental Protection Agency, Research Triangle Park, NC 27711, USA

**Keywords:** mixture risk estimation, independent action, toxicological interaction, hepatotoxicity, linear contrasts, Scheffé confidence interval, predictive computational toxicology

## Abstract

In this study, proportional response addition (Prop-RA), a model for predicting response from chemical mixture exposure, is demonstrated and evaluated by statistically analyzing data on all possible binary combinations of the four regulated trihalomethanes (THMs). These THMs were the subject of a multipurpose toxicology study specifically designed to evaluate Prop-RA. The experimental design used a set of doses common to all components and mixtures, providing hepatotoxicity data on the four single THMs and the binary combinations. In Prop-RA, the contribution of each component to mixture toxicity is proportional to its fraction in the mixture based on its response at the total mixture dose. The primary analysis consisted of 160 evaluations. Statistically significant departures from the Prop-RA prediction were found for seven evaluations, with three predications that were greater than and four that were less than the predicted response; interaction magnitudes (n-fold difference in response vs. prediction) ranged from 1.3 to 1.4 for the former and 2.6 to 3.8 for the latter. These predictions support the idea that Prop-RA works best with chemicals where the effective dose ranges overlap. Prop-RA does not assume the similarity of toxic action or independence, but it can be applied to a mixture of components that affect the same organ/system, with perhaps unknown toxic modes of action.

## 1. Introduction

Risk assessment of chemical mixtures is often based on dose–response properties of the chemical components because of the infeasibility of testing every mixture of concern. The formulas used in these mixture risk assessments have been widely studied, and usually have involved some version of additivity of component doses or responses, where “dose” here can have units of mass (e.g., mg chemical per kg body weight) or concentration (e.g., mmol per kg) and “response” can be the percent survival in a dose group or a specific effect measurement (e.g., enzyme activity). Example publications include those on the development and application of component-based approaches [1,2], determinations of additive joint toxic action and departures from additivity [3,4], and evaluations of additivity when multiple toxic mechanisms are present [5,6]. For routine risk assessment, biologically based procedures are expensive, information-intensive, and time-consuming. Consequently, dose–response evaluations of mixtures for use in risk assessment often involve simple component-based formulas.

In this research, scientists from several organizations developed a toxicology study protocol to investigate the accuracy of three specific component-based formulas under the leadership of the U.S. Environmental Protection Agency (U.S. EPA) [7]. The study was designed to evaluate proportional response addition (Prop-RA), a concept being evaluated by the U.S. EPA; the established interaction-based hazard index approach [8]; and statistical methods for detecting departures from dose-additivity (e.g., [9]). The study used CD-1 female mice and identified liver and kidney endpoints as the critical endpoints for four trihalomethanes (THMs), which are drinking water disinfection byproducts. This publication reports the results of evaluating Prop-RA; other publications are planned to present results for the other two projects.

The concept of dose addition, often used in mixture risk assessments, presumes some type of toxicologic similarity of the mixture components [10]. Support for this assumption can vary along a continuum of toxicologic evidence. The strongest support is an understanding of the common underlying molecular mechanism of action that leads to an adverse effect. As this complex information is rarely available, various less stringent levels of similarity, such as having a common mode of action (set of key toxicologic events) or merely sharing common adverse outcomes and pathways, may be used. In the absence of more detailed information, the simplest evidence of similarity that may be used for regulatory risk assessment is the components having the same target organ/system or the same syndrome of effects [8,11,12]. To avoid misinterpretations, we use “toxicological similarity” as a generic term to refer to any level of commonality along the continuum of toxicologic complexity.

In mixture risk assessment approaches, dose addition is implemented by combining component dose–response information with exposure data, again assuming some form of toxicologic similarity across components. Among the most commonly used dose-addition-based risk assessment approaches for mixtures is the hazard index (HI) [1,13,14,15,16,17]. More sophisticated forms of dose-addition use statistical modeling of the data, such as linear combinations of component doses and exposures, or nonlinear modeling techniques to describe mixture risk [18,19]. However, the form of dose addition based on Finney [20] also assumes congruent dose–response curves for the mixture components, while other formulations, such as that of Berenbaum [21], place no requirements on dose–response curve shapes. A second common formula for mixture risk is response addition (often called independent action). Its common implementation requires information on response frequency (e.g., fraction of a dose group showing toxic effects) and produces risk estimates as probabilities of a specified common effect. The main assumption for response addition is toxicologic independence of the components, meaning that they can individually affect a common target organ/system or cause the same type of toxicities; however, they do not share a common mechanism/mode of action nor a common adverse outcome pathway and the toxicity of one component does not impact the toxicity of any other component (e.g., all components cause cancer but may cause different types of cancer or cause cancer in the same organ but by different toxic mechanisms) [8,22]. This assumption allows the simple summing of conditional component response rates (per the statistical formula for independent events) to predict the mixture response [16]. Hybrid approaches have also been developed for more dissimilar mixtures, where both dose addition and response addition contribute to the mixture response estimate [23,24]. The standard HI and response addition formulas do not incorporate information on toxicological interactions between components. The interaction-based HI, as used by the U.S. Agency for Toxic Substances and Disease Registry (ATSDR) and as used by the U.S. EPA, incorporates evidence of binary interactions; its use is based on small to moderate departures from dose addition [1,13,25].

Because of the assumptions and data requirements of the methods described above, the appropriateness of their application to a given chemical mixture is often unclear. The accuracy and precision of these simple risk assessment methods, i.e., the HI, response addition, and interaction-based HI, have only been evaluated for a few mixtures. More advanced methods have been investigated for predicting mixture response, such as statistical modeling of component and mixture data for consistency with dose additivity [4,19], toxicodynamic models of interactions [26,27], and physiologically based pharmacokinetic modeling of joint toxic action [28,29]. These approaches, however, are hampered for general use by a lack of adequate dose–response data for many component chemicals and face challenges for the implementation of complex methods in many risk assessment situations. Thus, new tools for practical evaluation of joint toxic action are needed for risk assessment applications. In this study, we evaluate in detail a proposed component-based formula [30,31], one we call proportional response addition (Prop-RA). In Prop-RA, the predicted mixture response is the weighted average of the component responses at the total mixture dose, where each weight is that component’s dose fraction in the mixture. Chen, Heflich, and Hass [30] introduced the formula for Prop-RA by beginning with the classical mixture design (constant total amount, only the component fractions are changed), and then offering a variant where the total dose is also included. In related studies on mixture design, such “mixture-amount” models are often used to examine whether changes in total amount (e.g., dose of the chemical mixture) impact the “blending properties” of the components [32,33,34]. In Chen, Heflich, and Hass [30], the Prop-RA formula is applied to revertants in a salmonella assay, with no assumptions made regarding toxicologic similarity or independence. A toxicological rationale for Prop-RA has not been located in the open literature, thus, for this research it is considered to be a completely empirical approach.

The purpose of this present study is to demonstrate and critique the Prop-RA method using data on binary combinations of the four THMs. The purpose is not to compute an optimal dose–response model for THM mixtures, nor is it to develop a proposed THM mixture risk assessment. The specific goal is to identify and demonstrate the decision and calculation steps that are to be followed when the Prop-RA formula is applied in a component-based mixture risk estimation. The Prop-RA formula does not rely on having component dose–response models that are mathematically similar; thus, the single chemical dose–response evaluations are performed separately.

We first demonstrate the Prop-RA approach with the six possible binary combinations of four THMs, bromodichloromethane (BDCM), chlorodibromomethane (CDBM), bromoform (CHBr_3_), and chloroform (CHCl_3_), using molar fractions for the weights in the formula. We then recast the formula presented by Chen et al. [30] as a linear contrast to evaluate statistically significant departures from additivity, where additivity is defined by the Prop-RA formula. We also explore the influence of dose units (mmol/kg vs. mg/kg) on the formula predictions. Finally, we discuss possible underlying assumptions and constraints for general application to chemical mixtures from toxicological, statistical, and experimental design perspectives. Results showing a significant departure of the data from the Prop-RA prediction could indicate toxicological interactions or random deviations. Toxicological interpretation of such results is beyond the scope of this research. The goal of this effort is the development of a better understanding of the potential utility of the Prop-RA approach for risk assessment of chemical mixtures and possible research directions for its improvement.

## 2. Materials and Methods

The THM data were developed as part of a multipurpose toxicology study explicitly designed to allow the investigation of different approaches for addressing the joint toxicity of the four THMs. The experimental design is described below. Further details on experimental design and methodology can be found in Teuschler et al. (2000) [7].

### 2.1. Animals and Husbandry

Female CD-1 mice were obtained from Charles River Laboratory (Raleigh, NC, USA) at ~60 days of age. The animals were used in a facility certified by the American Association for the Accreditation of Laboratory Animal Care, and procedures were approved by the Institutional Animal Care and Use Committee (LAPR number 97-02-012 dated 27 February 1995) (U.S. EPA, Research Triangle Park, NC, USA). The animal room was maintained at a temperature of 22 ± 2 °C, at a relative humidity of 50 ± 10%, and on a 12/12 h light/dark cycle (lights on at 06:00 a.m). Mice were housed in polycarbonate cages with heat-treated pine-shaving bedding. They were allowed to acclimate to the animal facility for a minimum of 3 days before dosing; Prolab^®^ RMH 3000 (Land O’Lakes, Inc., Arden Hills, MN, USA) feed and tap water were allowed ad libitum throughout the duration of the experiment.

### 2.2. Chemicals in the Mixtures Used for Evaluation of Prop-RA

The test chemicals BDCM, CDBM, CHCl_3_, and CHBr_3_ were obtained from Sigma Aldrich Chemical Company (St. Louis, MO, USA). The supplier certified their purity as greater than 98%. The four THMs in this study are structurally similar and considered toxicologically similar. Their log p values, ratios of solubility in fat compared to water, are similar, ranging from 1.52–1.79, showing that all four are readily eliminated from the body (source: chemspi-der.com, accessed on 5 March 2024). They all have shown dose-related effects on the liver and on three serum enzymes indicative of hepatic injury: sorbitol dehydrogenase (SDH), alanine aminotransferase (ALT), and aspartate aminotransferase (AST). Three THMs are brominated and some results show brominated THMs to be more toxic than chlorinated THMs. For example, one acute study [35] found that BDCM caused significantly greater levels of serum hepatotoxicity markers than CHCl_3_ at 48 h post-exposure. They proposed that hepatotoxic potency differences between BDCM and CHCl_3_ “may be due to pharmacokinetic dissimilarities such as greater metabolism of BDCM to reactive metabolites or more extensive partitioning of BDCM into kidneys and fat depots, resulting in prolonged target tissue exposure”. The four THMs evaluated in this study co-occur in finished drinking water and have dose–response data on the same endpoints (e.g., hepatotoxicity); thus, they were considered appropriate for a component-based mixture evaluation, such as Prop-RA.

### 2.3. Experimental Design

Each THM combination was assessed in a separate experiment as described in Table 1. Eight different experiments were conducted. In all experiments, mice were exposed to one of the following eight binary combinations:

CHCl_3_:BDCM

CHCl_3_:CHBr_3_

BDCM:CHBr_3_

BDCM:CDBM

BDCM:CDBM (replicate)

CHCl_3_:CDBM

CHCl_3_:BDCM (replicate)

CDBM:CHBr_3_
EXPOSURES
Each THM combination was assessed in a separate experiment.Female CD-1 mice (outbred stock) were used as the animal model. Animals were assigned to treatment groups to ensure no statistically significant difference in body weight between treatment groups at the beginning of the experiment.A total of 14 days of daily dosing (dosing each day between 8 a.m. and noon).Dosing solutions were made fresh daily in "gas-tight" vials, immediately prior to dosing.An aqueous vehicle was used (10% Alkamuls EL-620, Rhodia Inc., Cranbury, N.J., USA, also known as Emulphor) to avoid the confounding effects of corn oil vehicle.Gavage volume was held constant at 10 mL/kg to avoid confounding by varying gavage volumes.Dose metric: administered chemical—mmol/kg/day.Hepatotoxicity was assessed on the morning following the 14th day of dosing by serum indicators and histopathology.Each experiment had 12 dose groups with 12 animals per group. These were as follows:A total of 1 aqueous vehicle control group.A total of 3 dose levels of THM “A” alone (0.1, 1.0, and 3.0 mmol/kg/day).A total of 3 dose levels of THM “B” alone (0.1, 1.0, and 3.0 mmol/kg/day.A total of 3 dose levels of the binary combination of A:B at a 1:1 mixing ratio (0.1, 1.0, and 3.0 mmol/kg/day). The 1:1 mixing ratio was present in all binary experiments.A total of 2 dose levels of the binary combination of A:B at a mixing ratio based on the concentrations of the THMs in drinking water (1.0 and 3.0 mmol/kg/day).This “environmentally relevant” mixing ratio was different for each binary combination.The environmentally relevant mixing ratios were as follows:

CHCl_3_:BDC2.7:1CHCl_3_:CDBM6.5:1CHCl_3_:CHBr_3_65:1BDCM:CDBM2.4:1BDCM:CHBr_3_24:1CDBM:CHBr_3_10:1

ENDPOINTS
Body weightSeveral expressions of body weight were examined.
(1)Weight gain over the course of the study. This was calculated as the weight on the first day of dosing (g) subtracted from the weight on the day of termination (g) as determined in the animal room.(2)Body weight (g) when the mice were terminated on day 15 (the day after the 14th day of dosing). This was SACWT (g).MortalityThe number of mice that did not survive the experiment was determined by counting the number of dead mice in each dose group.Organ weightsThe weights of the liver and the kidney were analyzed as relative organ weight [(weight of the organ (g)/SACWT (g)) × 100] (%). Our experience in general has been that relative liver weight (PCLIV) is more "sensitive" than absolute liver weight to the effects of these types of chemicals.Serum enzymes indicative of hepatotoxicityThe principal serum enzymes were ALT (IU/l, as international units per liter), AST (IU/l), and SDH (IU/l). Also, BUN (blood urea nitrogen), CREA (creatinine), and BUNCREA (BUN divided by CREA) were included because they are indicative of renal damage and they were verified at the same time as the other serum indicators.

### 2.4. Prop-RA Formula

Prop-RA, as a component-based approach, provides a prediction of the mixture response using only dose–response information on the mixture’s component chemicals. The formula for Prop-RA uses the fractions of each component in the mixture, the total dose of the mixture, and the response each component has when administered alone at that same total dose. For the individual component doses *(d_j_*), fractions (*π_j_*), and responses (*y_j_*) for components *j =* 1, …, *J*, we define total dose *D* as *D = d*_1_
*+ d*_2_
*+ … + d_J_*, where the response is any measure of effect or toxicity (e.g., percent of dose group affected, change in serum enzyme level). The Prop-RA formula for a binary mixture (so *J* = 2) is the weighted average:(1)$$y_{MIX}(d_{1},d_{2})=\pi_{1}y_{1}(D)+\pi_{2}y_{2}(D).$$

Here, “additivity” refers to this concept and formula, unless otherwise specified.

To keep the notation simple, the statistical models are presented without the usual datum subscripting, so that the subscripts in Equation (1) represent the chemical components. Each component dose is the total dose of the mixture multiplied by the component’s fraction in the mixture, so the Prop-RA mixture model can then be rewritten without explicitly showing the component doses. For the binary mixture in Equation (1), the left-hand side can be recast as:(2)$$y_{MIX}(d_{1},d_{2})=y_{MIX}(\pi_{1}D,\pi_{2}D)=y_{MIX}(D,\pi_{1})$$
where *π*_2_ is not needed on the right side because the fractions sum to one so *π*_2_ = 1 − *π*_1_. Then Equation (1) using notation from Equation (2) generalizes for a mixture of *J* components to become:(3)$$y_{MIX}(D,\pi_{1},\pi_{2},\ldots ,\pi_{J-1})=\pi_{1}y_{1}(D)+\pi_{2}y_{2}(D)+\ldots +\left(1-\sum_{j=1}^{J-1}\pi_{j}\right)y_{J}(D).$$

Note that the last fraction on the right-hand side of Equation (3) is determined by the other fractions, as shown, so the left-hand side need not include *π_J_* in the list of arguments. Because the fractions must sum to 1, the Prop-RA formula is then a weighted average of the responses expected from exposure to the same amount (*D*) of each of the individual components, where the weights are the relative fractions of the individual components in the mixture. Thus, if the mixture’s total dose is fixed, but the component doses are changed so that the dose of chemical 2 is doubled, say, from 10% to 20% of the mixture dose, the contribution of chemical 2 to the mixture toxicity will also double. To keep the total dose at *D*, the fraction (and thus contribution) of at least one other chemical must decrease. From Equation (3), we see that the predicted contribution of each component depends on the component’s fraction and on the single chemical response at a total dose (*D*), but not on the single chemical response at its actual dose in the mixture. The results are then quite different from the classical response addition defined by independent action.

For data on a fixed ratio ray (fractions are constant, total dosage varies), Equation (3) simplifies to a formula based only on D, which is the mixture total dose itself:(4)$$y_{MIX}(D)=\pi_{1}y_{1}(D)+\pi_{2}y_{2}(D)+\ldots +\pi_{J}y_{J}(D).$$

Under dose additivity, as presented by Finney [20], the components and the mixture are ideally described by the same dose–response model, and for a ray design, the mixture prediction formula has that same model structure [19]. With Prop-RA, there is no similar constraint on the component dose–response models. In this respect, Prop-RA is similar to both dose additivity as presented by Berenbaum [21] and to classical response addition; with Prop-RA each component dose–response model can be different, so the mixture prediction function should retain characteristics of all the separate component dose–response functions. Note that, from Equation (3), if all components are tested at the desired total dose *D* for the prediction, then the estimated mean mixture response can be a simple calculation using the experimental dose group means of the components, and no dose–response modeling is needed.

For example, consider the following dose group data for the single chemicals, BDCM and CDBM, with total dosage *D* = 3.0 mmol/kg and response as relative liver weight (%). The fraction is the molar dose fraction of each chemical in the corresponding binary mixture of these two chemicals.
ChemicalFractionRelative Liver Weight (%)BDCM0.7068.78CDBM0.2945.86

The predicted relative liver weight (%) for the binary mixture of BDCM and CDBM at *D* = 3.0 mmol/kg using the Prop-RA formula in Equation (1) with the single chemical dataset results is:(5)$$\begin{array}{cc} y_{MIX}(3.0) & =(0.706\times 8.78)+(0.294\times 5.86)\\ & =7.92 \end{array}.$$

From Equation (3), as mentioned above, the Prop-RA model requires estimated response values for each component at the same dose level, *D*. If one of the mixture’s total dose levels, say *D*_1_, is much higher than the highest dosage in the data for one of the components, then that component’s estimated response at dosage *D*_1_ must be extrapolated using a dose–response model for that component. When components have substantially different potencies, that extrapolation can be extreme or even impossible (for details and an example, see Section 4.2 “Complications of Total Dose in the Prop-RA Formula”). The THM study avoids such extrapolation by having the mixture and all four components tested at the same dose levels. For example, the binary mixture study of CHCl_3_:BDCM included three mixture dosages (0.1, 1.0, and 3.0 mmol/kg/day), with each component tested at those same three levels, i.e., it has a design with a common dose level across the components and the mixture (Figure 1). Each of the binary mixture experiments followed a similar concept, so no extrapolation was necessary. In fact, because each component was tested at the same dose levels as the mixture’s total dose, the single chemical data could be used directly in the Prop-RA formula to calculate a predicted mean response for the mixtures and then used to compare those predicted values with the observed means from the mixture data.

### 2.5. Methods for Modeling the Data

Toxicological rationales for Prop-RA have not been proposed; therefore, this study is considered to be a completely empirical approach. The statistical methods applied here determine the significance of deviations from Prop-RA. As such, the results cannot verify that Prop-RA is the correct model but can only establish consistency between the data and the Prop-RA formula.

Prop-RA involves a linear combination of component responses, where the coefficients are the component fractions in the mixture. Based on Equation (5), for a fixed dosage, *D*, the observed mixture mean [left side of Equation (6)] can be compared with the predicted mixture mean [right side of Equation (6)], with the latter calculated as the weighted average of the observed component means (each at total dose *D*) with weights of the component fractions.
(6)$${\bar{y}}_{MIX}(D,\pi_{1},\pi_{2},\ldots ,\pi_{J})=\pi_{1}{\bar{y}}_{1}(D)+\pi_{2}{\bar{y}}_{2}(D)+\ldots +\pi_{J}{\bar{y}}_{J}(D)$$

If the mixture mean is subscripted by *J* + 1 and is moved to the right-hand side, and we use population means, *µ_i_* (unknown but theoretically correct), instead of sample means (used for estimation in Equation (12) below), we obtain Equation (7), where *L* is the difference between the prediction and the observation. Here the parameters *c_j_* are the fractions in Equation (6) for *j* = 1 to *J*, and *c_J+_*_1_ = −1.
(7)$$L=\sum_{j=1}^{J+1}c_{j}\mu_{j}$$

We see from Equation (7) that *L* is a linear contrast in the form of a linear combination of two or more means such that the coefficients of the linear combination sum to zero [36]:(8)$$\sum_{j=1}^{J+1}c_{j}=0.$$

Linear contrasts are commonly employed in statistical inference for comparing several means, and the concept is applicable here for testing consistency with Prop-RA. Suppose, for example, we have three means denoted by *μ*_1_, *μ*_2_, and *μ*_3_ and are interested in evaluating whether *μ*_3_ is equal to the average value of the other two means. We could calculate 0.5 *μ*_1_ + 0.5 *μ*_2_ − *μ*_3_ (note that the coefficients 0.5, 0.5, and −1 sum to zero) and observe whether the result is zero. This is equivalent to a binary mixture example, where *μ*_3_ is the mixture mean.

If a mixture of *J* components is represented by their ratios, i.e., as *A*_1_:*A*_2_:*A*_3_: ….: *A_J_*, then the Prop-RA model indicates that the expected response at a fixed dose for the mixture would be *μ_J_*_+1_ where:(9)$$\mu_{J+1}=\frac{A_{1}}{A_{1}+A_{2}+\ldots +A_{J}}\mu_{1}+\frac{A_{2}}{A_{1}+A_{2}+\ldots +A_{J}}\mu_{2}+\ldots +\frac{A_{J}}{A_{1}+A_{2}+\ldots +A_{J}}\mu_{J}.$$

For example, if there are two components in the mixture and the ratios are 1:3, this would be:(10)$$\mu_{MIX}=\frac{1}{1+3}\mu_{1}+\frac{3}{1+3}\mu_{2}=0.25\mu_{1}+0.75\mu_{2}.$$

So, if we construct a contrast where the coefficients are:(11)$$c_{j}=\frac{A_{j}}{A_{1}+A_{2}+\ldots +A_{J}}$$
for *j* = 1, 2, …, *J* and *c_J_*_+1_ = −1, we again have a linear combination that compares *μ_J_*_+1_, the response to the mixture, with the predicted response under Prop-RA. If the Prop-RA model holds, then the contrast will be zero. Note that Equations (9) and (11) are mathematically identical to their counterparts using fractions instead of ratios, e.g., *c_j_* in Equation (11) equals *π_j_* in the right-hand side of Equation (6).

The evaluation of consistency with Prop-RA is then straightforward: estimate the contrast with a confidence interval and observe whether zero falls in that interval. If the interval falls entirely to the right of zero (i.e., both the lower limit and the upper limit are positive) a result less than expected under Prop-RA is indicated and if the interval falls entirely to the left of zero a result greater than expected under Prop-RA is indicated. Note that *L* is a combination of effect means; its numerical scale is that of the response being measured. The confidence limits for *L* should then not only reflect the standard deviation of the effect measure but also be numerically higher for those endpoints with numerically higher measures; in this case, the serum enzymes are indicative of hepatotoxicity.

To obtain a confidence interval for *L*, we start with an unbiased estimate of *L* by substituting sample estimates for the unknown population means, *μ_j_*, in Equation (7) to obtain:(12)$$\hat{L}=\sum_{j=1}^{J+1}c_{j}{\bar{y}}_{j}.$$

Then, if we assume homogeneity of variance (HOV), an unbiased estimate of the variance of the estimate is given by:(13)$$VAR=MSE\times \sum_{j=1}^{J+1}\frac{c_{j}^{2}}{n_{j}}$$
where *MSE* is the usual pooled weighted average of the individual sample variances using degrees of freedom as the weights, and *n_j_* is the sample size for the *j*^th^ chemical.

If only one confidence interval is to be calculated, it can be based on the Student’s t-distribution as:(14)$$\hat{L}\pm t\left(1-\alpha /2,N-(J+1)\right)\times s(\hat{L})$$
where (1 − *α*) × 100% is the desired confidence level, *s* is the standard deviation, and *N* is the total sample size [36].

When several confidence intervals are to be constructed from the same data, it is preferable to use a method that controls the multiple evaluation error, such as the Scheffé or Bonferroni method [36]. In general, the Scheffé method will yield narrower confidence intervals when a large number of comparisons are made. The Scheffé method is similar to the Student’s t method described above except that the tabulated *t* in Equation (14) is replaced by a function of the F-distribution with degrees of freedom *J* and *N* – *J* − *1*. The interval is then:(15)$$\hat{L}\pm \sqrt{J\times F(1-\alpha ,J,N-J-1)}\times s(\hat{L}).$$

Scheffé’s method tests all possible contrasts at the same time to determine if at least one is significantly different from zero. The Scheffé approach does not just consider pairwise differences but rather is applied to the set of estimates of all possible contrasts. Technically, there are an infinite number of contrasts; thus, the simultaneous confidence coefficient is 1 − *α*. As mentioned above, the Scheffé method will yield narrower confidence intervals when a large number of comparisons are made. The overall type I error rate for this study using Scheffé’s method is *α* = 0.05 [37].

Each linear contrast is constructed for a given experiment, dose, ratio, and endpoint; for this combination, HOV is assumed across the two single chemicals and the binary mixture. We used the O’Brien test for HOV at *α* = 0.05 to check that assumption for the use of Equation (13). The O’Brien test was chosen based on published research showing that it is the only HOV method that has adequate control of type I error rates for average sample sizes < 10; it is not sensitive to normality assumptions of the data; and it controls for type I error rate across all population shapes [38]. In general, it is not sensitive to skewed data, although this can sometimes be an issue with small sample sizes.

HOV is an important assumption because calculating the pooled variance assumes the three estimates of variability have been drawn from the same population. If the three sample variances are from different populations, then pooling them could lead to erroneous results. For example, if one variance is larger than the other two, this could increase the size of the pooled standard deviation and subsequently increase the size of the confidence interval. This results in decreased statistical power, making it harder to reject the null hypothesis that Prop-RA is a good model for the data. Conversely, if one variance is smaller than the other two, a tighter confidence would be produced, resulting in increased statistical power that could lead to erroneously rejecting the null hypothesis. Thus, choosing the datasets that best meet the HOV assumption will yield the most scientifically correct results.

In our analysis, if the HOV assumption was not met for the original data, we transformed the original data using a common variance stabilizing transformation, i.e., by taking the log10 of the response data [36]. Where transformation was indicated, the results based on the transformed data are presented. To understand the influence of transformation on the Prop-RA results, analyses were run with the original data (see Section 3.4).

### 2.6. Evaluating the Influence of Dose Unit on Prop-RA Prediction

The mixing ratios (and thus component fractions) can be numerically different when the daily dose unit is changed, e.g., from mmol/kg to mg/kg, which would likely change the predicted mixture response. We investigated the extent of that influence of dose unit selection by using the exposure data for the binary combination CHCl_3_:CHBr_3_ at a 1:1 molar mixing ratio. We then calculated the component fractions and total mixture dose if the component doses were converted to mg/kg. Last, we converted this altered total dose back into units of mmol/kg for comparison with the active dose range for those two component chemicals.

## 3. Results

HOV testing results were helpful in ascertaining the appropriateness of applying our method for determining departures from Prop-RA to the relative liver weight and serum enzyme data for the THMs (full results for each of the 160 evaluations conducted and from the HOV testing process are shown in Appendix A). For relative liver weight data, the variance was fairly consistent across the components and mixture (example in Figure 2). HOV testing showed general consistency (i.e., no significant differences) across all but two experiments for dose/ratio/endpoint combinations across the two single chemicals and the binary mixture. Thus, for relative liver weight, we chose to use the original untransformed data for testing departures from Prop-RA. In general, HOV testing results for the serum enzyme data were less consistent than for the relative liver weight data. To support more closely the assumption of constant variance, the enzyme response data were transformed using a log10 conversion [36]. Variances were more homogeneous with the log10 transformation based on O’Brien test results and visual inspection of changes in the similarities of the variances (Figure 3). Thus, relative liver weight (PcLiv, as percent body weight) and the three log-transformed serum enzyme levels (ALT, AST, and SDH) were chosen to be appropriate endpoints for demonstrating the calculations used to determine departures from Prop-RA. The first five subsections show results using mmol/kg as the dose unit. The final subsection presents the influence of the dose unit on the predicted mixture response.

For each of the four endpoints (PcLiv, ALT, AST, and SDH) evaluated within each binary combination experiment, five “departure from Prop-RA” comparisons were conducted, one for each of the five dosage/mixing ratios tested (see example in Table 2). Thus, 20 Prop-RA evaluations (five for relative liver weight and 15 for serum enzymes) were conducted for each of the six binary combinations studied and the two study replicates. Overall, a total of 160 evaluations were conducted (four endpoints times five dose levels times eight experiments) of which 120 were for the six binary combinations and 40 evaluations were for the replicate experiments for two of the binary combinations.

In the remainder of this section, example calculations are shown in detail for relative liver weight with a binary mixture. Then, results are presented that detail the departures from Prop-RA that were found. Table 3 summarizes these results for all the endpoints. The rows showing 95% Scheffé confidence intervals in the last column represent the seven statistically significant departures from Prop-RA that resulted from the 160 comparisons. The other rows describe the endpoints, dosages, and ratios whose data were consistent with Prop-RA. Finally, summary results about consistency with Prop-RA are presented for all binary mixtures and endpoints; complete results are presented in Appendix A.

### 3.1. Example Numerical Results for Relative Liver Weight for BDCM:CDBM Mixtures

As an example of the calculations we are using, consider the relative liver weight results for the mixing ratio of 2.4:1 for BDCM:CDBM (i.e., fractions of 0.706 and 0.294) at the dosage of 1.0 mmol/kg/d (fourth row of Table 2). Under Prop-RA, with BDCM and CDBM means of 5.94% and 6.09%, respectively, and the mixture mean of 5.78%, the predicted mixture mean from Equation (3) and linear contrast from Equation (12) for PcLiv are calculated as:Predicted mean: (0.706 × 5.94) + (0.294 × 6.09) = 5.98.
Linear contrast (*L*): (0.706 × 5.94) + (0.294 × 6.09) − 5.78 = 0.204

By reversing the calculation, the predicted mean percent liver weight can be calculated from the observed mean and the linear contrast: 5.78 + 0.204 = 5.98.

The 95% Scheffé confidence interval for the linear contrast is calculated using Equation (14) above, with an overall α level of 0.05. For this example, the following values were obtained:*N* = *N_BDCM_* + *N_CDBM_* + *N_MI_*_X_ = 20,
*J* = no. of components = 2

Calculations of the Scheffé interval are then as follows:

Degrees of freedom for the F test: *J* = 2 and *N* – *J* − *1* = (20 − 2 − 1) = 17

Value of the F statistic: *F*(0.95, 2, 17) = 3.59

Mean square error: *MSE* = [(6 − 1) × (0.44)^2^ + (7 − 1) × (0.48)^2^ +(7 − 1) × (0.30)^2^] ÷ 17 = 0.17,

Variance (Equation (13)): *VAR* = *MSE* × [(0.706^2^ ÷ 6) + (0.294^2^ ÷ 7) + (−1^2^ ÷ 7)] = 0.0408

Standard deviation: *s* = (*VAR*)^0.5^ = (0.0408)^0.5^ = 0.202

Confidence interval from Equation (15):(16)$$\begin{array}{cc} CI & =0.204\pm \left(\sqrt{2\times 3.59}\times 0.202\right)\\ & =\left(-0.34,0.74\right) \end{array}.$$

Because the confidence interval, (−0.34, 0.74), includes zero, the component and mixture data for relative liver weight are consistent with Prop-RA, i.e., the observed response of 5.78% is not statistically different from the prediction of 5.98%.

### 3.2. Relative Liver Weight Numerical Results

For relative liver weight, the six binary THM mixtures (plus two replicates) were each compared for five dosage/ratio combinations, thus 40 comparisons, of which 10 were for the replicate studies (CHCl_3_:BDCM-rep and BDCM:CDBM-rep). For relative liver weight, one departure from Prop-RA was identified for the binary mixture BDCM:CDBM (details in Table 2 and Table 3); this result was not found in the replicate study of this mixture. The results shown in Table 3 for all the other experiments were consistent with predictions using Prop-RA for relative liver weight, i.e., the confidence limits for the linear contrast contained zero (full results for the 40 relative liver weight evaluations are shown in Appendix A).

The direction of non-additivity can be determined by comparing the predicted response with the observed mean, e.g., a larger observed mean relative liver weight indicates greater than additive joint toxic action. This is the case for the one significant relative liver weight result for the mixing ratio of 2.4:1 for BDCM:CDBM at a dosage of 3.0 mmol/kg/d (Table 2 and Table 3). The 95% confidence interval for the linear contrast does not contain zero (−4.11, −0.44), indicating a departure from Prop-RA. The linear contrast is the prediction minus the mixture means (7.92 − 10.19 = −2.27), so this significant negative contrast suggests a greater-than-additive response with respect to Prop-RA while significant positive contrasts suggest a less-than-additive response with respect to Prop-RA. Greater-than-additive interactions are typical of greater public health concern (than less-than-additive results), as the risk would be underestimated by the use of a risk assessment method based on an assumption of additivity.

### 3.3. Serum Enzyme Numerical Results

ALT, AST, and SDH data were evaluated for consistency with Prop-RA for the binary mixtures using the log10 transformation for each of the three enzymes; 120 comparisons were made, of which 30 were for the replicate analyses (CHCl_3_:BDCM-rep and BDCM:CDBM-rep). Table 3 shows that serum enzyme results from the binary mixtures for which statistically significant departures from Prop-RA were found, using the same steps described above for relative liver weight. For these endpoints, six departures from Prop-RA out of 120 evaluations were identified, with all other mixtures showing consistency with the formula (full results shown in Appendix A). The departure from Prop-RA shown for AST for the binary combination of CHCl_3_:BDCM at a dosage of 1.0 mmol/kg was not found in the replicate study of this mixture.

### 3.4. Alternative Analysis with Untransformed Serum Data

As a result of our HOV analysis, the data of ALT, AST, and SDH were analyzed using a log10 transformation. To examine the impact of that decision, we also conducted departures from Prop-RA comparisons using the original, untransformed datasets. Table 4 provides results that compare the original data (fifth column) with applying a log10 transformation to the serum enzyme data (sixth column). As an example of the data presented, row two gives the results for the binary combination CHCl_3_:CHBr_3_ at a 1:1 ratio and a dosage of 3.0 mmol/kg/day. The designation of “PRA” in column five indicates that the results were consistent with Prop-RA. The Scheffe confidence interval in column six indicates a departure from Prop-RA when the interval does not include zero. The *p*-values below the comparison results are the significance levels of the HOV analysis. Using the O’Brien test (*p* < 0.05 significance level), HOV was rejected for both the original data (*p* = 0.003) and log10-transformed data (*p* = 0.04). However, a departure from Prop-RA was found for the log10-transformed data, but not for the original data, which is consistent with seeing a higher HOV *p*-value for the transformed data.

A general comparison of the log10 transformation results with results obtained with the untransformed data shows that only three departures from Prop-RA (out of 120 evaluations) were found for the serum enzymes using the untransformed data; two of these were included within the six detected using the log10-transformed data. For the other four departures from Prop-RA using the log10-transformed data (rows 1, 2, 3, and 10, Table 4), the HOV testing results were about the same or improved. With the log transformation, there were additional detections of departures from Prop-RA. For CHCl_3_:BDCM, we note that the consistency with Prop-RA was also seen in the replicate experiment when using the untransformed (i.e., original) data but that departure from Prop-RA seen for the log-transformed data was not observed in the replicate study.

### 3.5. Summary of Results with Dose as mmol/kg

Table 5 summarizes the statistically significant departures from Prop-RA, for both relative liver weight and the serum enzymes. Only 7 out of 120 evaluations from the six original binary experiments showed departures from Prop-RA. There were no departures from Prop-RA identified for the 40 comparisons for the two replicate experiments (CHCl_3_:BDCM-rep and BDCM:CDBM-rep), so those studies are not shown in Table 6. Two departures from Prop-RA found in the original CHCl_3_:BDCM and BDCM:CHBr_3_ experiments (Table 5) were not detected in the replicates. It can be noted, however, that the other 38 comparisons for the original and replicate studies were the same, all showing consistency with Prop-RA.

Differences in response means (predicted–observed means) and the interaction magnitudes (n-fold change in response) are used to evaluate the strength of the Prop-RA predictions. Reproducibility of results is desirable to reinforce such interpretations but replicated mixture experiments are rare. Interaction magnitude here is calculated as the n-fold change in the measured endpoint at a fixed dose. The three smaller interaction magnitudes (1.3 to 1.4) are all associated with observed means larger than predicted, thus indicating greater-than-additive joint toxic action. These include AST at dosages of 0.1 and 1.0 mmol/kg/d for CHCl_3_:CDBM at a 1:1 ratio and CHCl_3_:BDCM at a 2.7:1 ratio, respectively, and relative liver weight at a dosage of 3.0 mmol/kg/d for CHCl_3_:CDBM at a 2.4:1 ratio. To date, Prop-RA is not a risk assessment method recommended for use by any regulatory agency and this study represents the first known effort to evaluate the usefulness of Prop-RA for mixture risk assessment. Under the hypothetical situation that a risk assessment was based on an assumption of fractional response additivity, these greater than Prop-RA interactions could mean the risks from exposure to the mixture are underestimated. The value of the interaction magnitude (in all cases less than 1.5) would provide useful information to the risk assessors. The four results with the largest interaction magnitudes (2.6 to 3.8) are all associated with observed means smaller than predicted by Prop-RA, indicating less than additive joint toxic action, so there is no concern in those cases about biological upper limits influencing the response measures [39]. All of these results are for the enzymes ALT, AST, and SDH (at the high dose of 3.0 mmol/kg/d) with two 1:1 mixing ratios and two environmentally relevant mixing ratios. It is also noteworthy that all four of these highest interaction magnitudes of 2.6 to 3.8 were for mixtures that contain the highly brominated THM and CHBr_3_, while none of the three smaller interaction magnitudes included CHBr_3_.

All of the comparison results are displayed in Figure 4 (relative liver weight, logALT) and Figure 5 (logAST and logSDH). Even with the log10 transform, the enzyme variances are clearly not as consistent as those for relative liver weight, as indicated by the larger deviations from the prediction that were not statistically significant. Of the 40 evaluations for each endpoint, relatively few showed a departure from additivity as defined by the Prop-RA formula. These four graphs, Figure 4a,b and Figure 5a,b, show a collection of dose-specific comparisons and should not be interpreted in terms of trends across the range of responses. It is better to view the graphs more as a summary or index of the results across the 40 evaluations for each endpoint and not necessarily an overall appraisal of the quality of the Prop-RA predictions across the range of these mixture studies.

### 3.6. Influence of Dose Unit on Prop-RA Prediction

Changing from the daily dose as mmol/kg to mg/kg altered the total dose magnitude and the component dose fractions (Table 6). Consider one of the original total doses, 1.0 mmol/kg, used with this binary mixture, and the mixing ratio of 1:1 (CHCl_3_:CHBr_3_). The first step is calculating the component doses (=fraction × total dose) and scaling them from mmol/kg to mg/kg by multiplying by the mol wt of 119.38 for CHCl_3_ and 252.73 for CHBr_3_. Then, the total mixture dose in mg/kg is the sum of those two scaled component values (Table 6, fourth column). The mg based component fractions are then calculated using the mg based total mixture dose; for CHCl_3_ it is 59.69/186.06 = 0.32 using values from the first row. Thus, changing units changed the CHCl_3_ fraction from 0.5 to 0.32. For the total dosage of 3 mmol/kg, the original 1:1 mixing ratio (with component doses of 1.5 mmol/kg each) is converted into component doses of 179.07 mg/kg for CHCl_3_ and 379.10 mg/kg for CHBr_3_. The corresponding total dose of 3 mmol/kg is converted to 558.17 mg/kg. As shown above, for CHCl_3_, its 0.5 fraction under mmol/kg is now 0.32 under mg/kg. The total dose at which CHCl_3_ would need an estimated response is much higher relative to the tested CHCl_3_ exposure, with the new total dose of 558.17 mg/kg converting to 4.68 mmol/kg of CHCl_3_. The highest tested CHCl_3_ dose was 3 mmol/kg.

## 4. Discussion

### 4.1. Conditions Related to the Application of the Prop-RA Formula

Component-based approaches for predicting mixture response usually place certain conditions or assumptions on the dose–response characteristics of the mixture and its components. For example, the classical response addition formula is supported by the assumption of toxicological independence and the dose addition formula is supported by the assumption of toxicologic similarity [1]. With Prop-RA, the component chemicals cannot be considered toxicologically independent because of the inclusion of the total dose in each component’s contribution to the formula. To see this, consider Equation (2) for a binary mixture with total dose *D* replaced by the sum of component doses. The direct contribution of the first chemical (*d*_1_) to the mixture toxicity is:(17)$$\pi_{1}y_{1}(D)=\frac{d_{1}}{d_{1}+d_{2}}y_{1}(d_{1}+d_{2})$$
which depends on the dose of the second chemical (*d*_2_). For Equation (17) to be calculated (or in general for each term in Equation (3)), there must be toxicity information (*y*), either estimated or measured, on the components at the same dose, *D*. We first assume that the components have some variation in potency so that they do not all have the same response at dose *D*. Consequently, each of the component chemicals needs its toxic response range to be associated with a dose range that includes *D*. While this dose range overlap might occur with any group of chemicals, chemicals that are roughly toxicologically similar would seem to have a greater chance at dose range overlap. That overlap is assured if there is quantitative consistency across components of the maximal dose causing toxic but not lethal effects.

The data evaluated here involve four THMs that have roughly similar dose–response profiles. All showed similar measures of hepatic dysfunction over the same dose range of 0.1–3.0 mmol/kg/day, and most of the predicted mixture responses were not statistically different from the measured mixture mean. The statistical approach described is easy to implement, particularly when the evaluation doses are actual doses tested with all components and with the mixture. In that case, the predicted mixture response can then be calculated without any dose–response modeling, and the comparison of predicted vs. observed can then be performed on a dose-by-dose basis.

### 4.2. Complications of Total Dose in the Prop-RA Formula

Regarding mixtures with quantitatively dissimilar chemicals and the need for extrapolation, the inclusion of the total dose in the formula can lead to calculation difficulties when applied to quantitatively dissimilar chemicals (different toxic potencies and non-overlapping effective dose ranges) because of the need for model extrapolation, which is possibly well beyond the highest dose of some of the component chemicals. For example, exposure to a weakly toxic chemical might have no detectable effect on mixture response at extremely high concentrations, whereas exposure to a structurally similar but highly toxic chemical might result in essentially 100% response at low concentrations. Testing the second chemical in the same assay at a moderate concentration could result in high mortality and thus not be useful for predicting a nonlethal toxic response. A prediction using Prop-RA for a mixture containing that second chemical, even at a low mixing ratio, would need to include its severe response as a predicted endpoint. The hypothetical example below, based on data for a toxicologically similar group of chemicals, shows how having non-overlapping dose ranges leads to undesirable extrapolation.

Let chemicals A-E be a group of toxicologically similar chemicals that include a wide range of relative potencies. In this hypothetical example, the range of tested doses illustrates the extreme differences in toxic potency for a given response of interest (Figure 6). For example, there is no dose range overlap between chemicals A and D. While 8000 µmol is a moderately toxic dose of chemical D, it is 320× the highest dose tested with chemical A, at which the response measure was essentially 100%. Let y_A_ denote the dose–response function for chemical A. The term analogous to Equation (17) for chemical A then needs an estimated response at that dose: y_A_(8000). Extrapolating the dose–response model for chemical A that is far beyond the data is extremely uncertain from both mathematical and toxicological perspectives, which strongly contraindicates the application of Prop-RA to mixtures of these two chemicals.

Concerning the dependence on dose units resulting in a change in mixing ratio, the mixture predictions based on Prop-RA depend on dose units if the units represent specific physical measures. There is no difficulty with a rescaling of the same concept, such as mass. For example, changing the dose from mmol/kg to mmol/pound results in the same shape dose–response curve (whether single chemical or mixture) because the change merely stretches or contracts the dose axis. The concern here is when switching to a functionally different dose measure, e.g., mass density (mg/kg) instead of molecular density (mmol/kg). Such a change has no impact on a single chemical’s dose–response curve because the result is merely an expansion or contraction of the dose axis by the molecular weight of the chemical. For dose addition, that change in dose unit likewise has no impact on the mixture response prediction [40]. This is because the product of the dose coefficient and dose in the dose addition formula is the same for different dose units: a smaller dose magnitude (e.g., with dose as mmol/kg instead of mg/kg) is balanced by a correspondingly larger coefficient. With Prop-RA, however, the situation is different. The inclusion of total dose in the formula causes the mixing ratios (and thus component fractions) to change when switching from mmol/kg to mg/kg, and thus each component’s response prediction changes as well because of the changed total dose.

These consequences are shown (Table 6) in terms of altered total dose and altered component fractions with a resulting need for extrapolation at the converted total dose, using the CHCl_3_:CHBr_3_ binary mixture experiment. The converted total dose values are different from any tested values, so will have to be estimated from a dose–response model. Such estimation can cause problems, such as the extrapolation mentioned above. As shown in Table 6, fifth column, at the high total dose (originally 3 mmol/kg, now 558.17 mg/kg), the corresponding total dose in original units for CHCl_3_ of 4.68 mmol/kg is well outside the tested range, which is 50% higher than the highest tested dose of 3 mmol/kg where some mortality was observed. When using such an extreme extrapolation in the Prop-RA formula, the prediction is unlikely to be viewed as a plausible mixture response estimate.

### 4.3. Improvements in the Prop-RA Evaluation

An improvement in this approach might be to compare dose–response models across the full tested dose range, instead of comparing dose groups. Model–model comparisons have been used and demonstrated with dose-additivity evaluations [41], including the similarity of both response means and variances [19]. Model comparisons by definition reflect general responses across the full dose range and so partly compensate for uneven dose group sizes, e.g., where some dose group comparisons showing consistency may do so due to poor statistical power to detect a departure, reflecting small sample sizes and high variances. Related improvements would include adjustment for multiple comparisons (across endpoints and dose groups) and direct modeling of variance as a function of dose or response.

Another improvement in understanding the Prop-RA formula is to develop theories of why total dose should influence the contribution of each component to the mixture toxicity. In the reference that triggered this present investigation [30], emphasis was placed on the importance of total dose, not just the component doses, but no supporting biological arguments were presented. As noted above, the numerical prediction changes based on the units of dose, so an interesting research direction is whether the dose level expressed as mmol/kg (molecular density) might better explain joint toxicity than the dose level expressed as mg/kg (mass density). One supporting argument is that toxicodynamics is often based on molecule–cell interaction (e.g., the number of molecules binding to a receptor). Until such biological theories are more completely formulated, the Prop-RA formula should be considered as a simple, empirical component-based alternative to the existing formulas of dose addition and response addition.

### 4.4. Interaction Magnitude as a Function of Predicted vs. Observed Response

In Table 5, we presented two metrics for evaluating interaction magnitude under Prop-RA: absolute difference in response predicted from the mixture components and the measured response from the mixture data, and the n-fold change in response, calculated as the ratio of these two values. While using measures of response to estimate interaction magnitude was considered in the initial U.S. EPA mixture risk guidelines [42], it contrasts with the most current U.S. EPA definition for the interaction-based hazard index [43] and with the similar ecotoxicity metric of the Model Deviation Ratio [44], where in both cases interaction magnitude is the n-fold change in effective exposure level (e.g., a ratio of ED10s). Such an exposure-based definition of interaction magnitude is more readily understandable in risk management terms, e.g., the n-fold change in clean-up goals. In contrast, the response-based interaction magnitude for Prop-RA has the advantage of being interpretable in terms of actual changes in the effect of concern.

## 5. Conclusions

The choice of a simple model for evaluating possible toxicological interactions or predicting mixture toxicity cannot yet be based on mechanistic grounds. Progress has been made for simple mixtures (few components) considering metabolic interactions with physiologically based toxicokinetic modeling techniques [45,46,47,48,49]; those biologically based approaches are resource-intensive, and thus are not suitable yet for routine risk assessment. The definition of dose addition adopted by the U.S. EPA (1986) carries an assumption of a shared mechanism/mode of action or adverse outcome pathway; this underlying toxicological concept motivates its formula and supports its application. Classical response addition also has an underlying toxicological concept, namely independence of toxicological modes of action.

Applying Prop-RA, as with any component-based formula, requires a common toxic endpoint or target organ/system across the components. In the case study presented here, the Prop-RA approach to evaluate departures from additivity was demonstrated with data from the binary combinations of the four regulated THMs. A total of 160 evaluations were conducted; 20 evaluations were conducted for each of the six binary combinations and for the two study replicates. Out of these 160 assessments of consistency with Prop-RA additivity, only three greater-than-additive responses were detected. Interpreting such results as indications of toxicological interaction vs. random variation would require further investigation. As presented by Chen et al. [30], the Prop-RA method does not require either an assumption of toxicological similarity or independence. The example calculations in this paper argue for an additional requirement of a similar range of active doses across components. The development of a toxicological explanation for the impact of total dose on each component’s contribution to the mixture toxicity is a particular research need that would support Prop-RA as an alternative mixture risk formula. The statistical methods applied here have provided a thorough evaluation of the consistency between the experimentally observed results and the results predicted by the Prop-RA model. In an additional evaluation of the Prop-RA model, future efforts will compare the results obtained here with Prop-RA with those obtained from the examination of these same data with a well-established dose-additivity model, thus representing an important step in understanding the public health relevance of the additivity results obtained by the Prop-RA method.

## Figures and Tables

**Figure 1 toxics-12-00240-f001:**
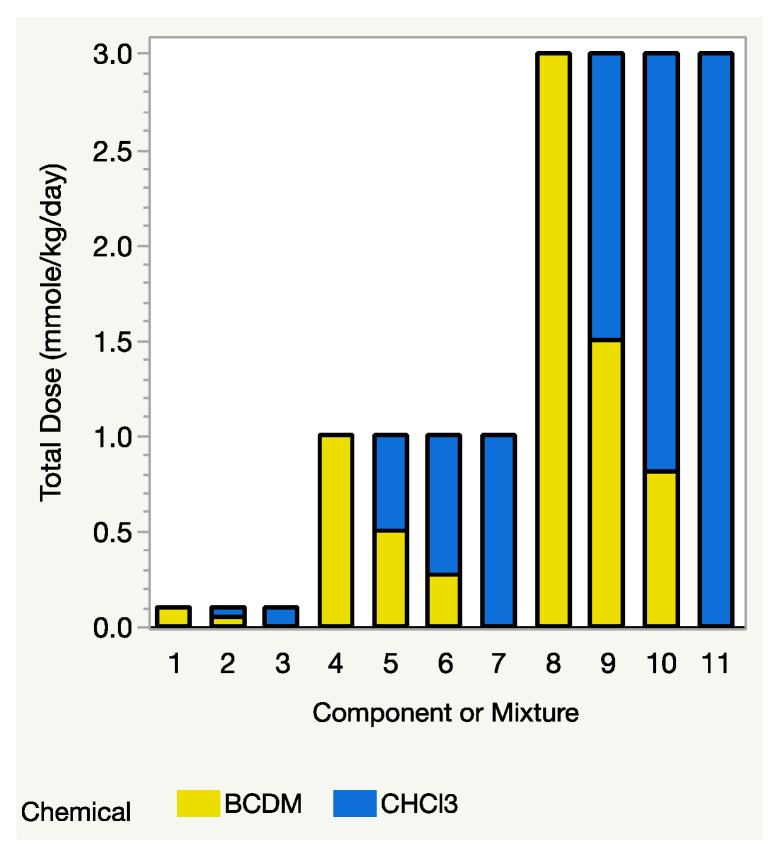
Design for evaluating Prop-RA using binary mixtures. Total dose = dosage of each component alone, or mixture dosage (sum of component dosages). Three total doses of 0.1, 1.0, and 3.0 mmol/kg/day were used with each component and with the 1:1 mixture. Only the higher dosages of 1.0 and 3.0 mmol/kg/day were used with the environmental mixture (2.7:1 for this example of CHCl_3_:BDCM). Comparing the height of each two-colored bar with the height of the adjacent bars shows that the mixture dosage is also one of the dosages for each of the component chemicals.

**Figure 2 toxics-12-00240-f002:**
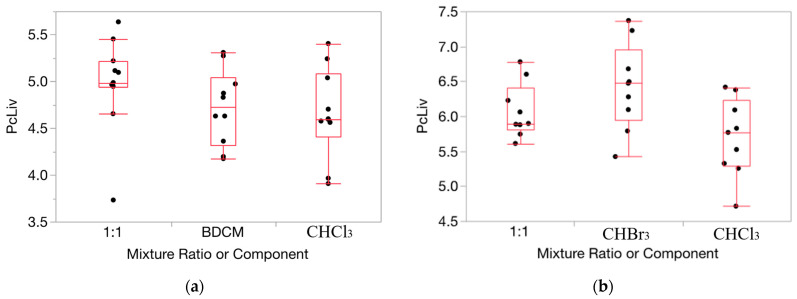
Data on percent relative liver weight (PcLiv) show fairly consistent variance over the components and mixture. (**a**): CHCl_3_, BDCM, and the mixture (1:1 ratio) at dosage = 0.1 mmol/kg/day. (**b**): CHCl_3_, CHBr_3_, and the mixture (1:1 ratio) at dosage = 1.0 mmol/kg/day. Note: the vertical dimension of each outlier box shows the interquartile range with the horizontal line in the box denoting the median; the dots are jittered data points.

**Figure 3 toxics-12-00240-f003:**
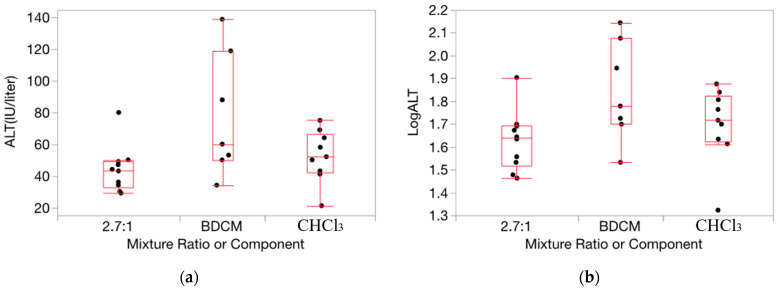
Data on ALT activity (IU/liter) for exposure to CHCl_3_, BDCM, and the mixture (2.7:1 ratio) at dosage = 1.0 mmol/kg/d from CHCl_3_:BDCM-rep, showing the stabilizing influence on variance by the log transform (boxes are more uniform in length). (**a**): Response as ALT. (**b**): Response as log10(ALT). Note: the vertical dimension of each outlier box shows the interquartile range with the horizontal line in the box denoting the median; the dots are jittered data points.

**Figure 4 toxics-12-00240-f004:**
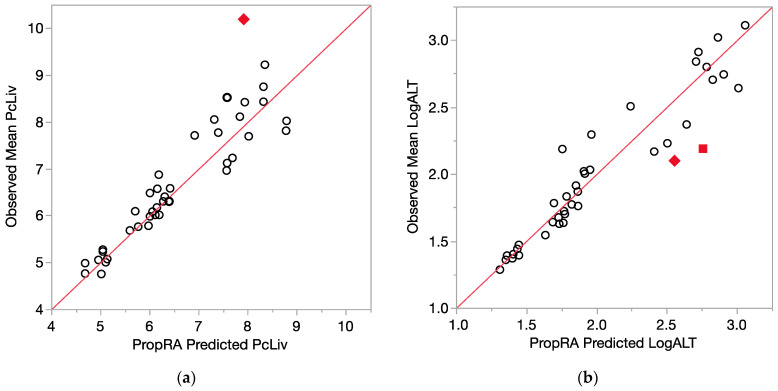
Consistency of data with the Prop-RA prediction for PcLiv and LogALT. Circles indicate concordance between data and Prop-RA prediction. Filled-in symbols denote statistically significant departure from the Prop-RA prediction for that dose–mixture combination. The red 45-degree line is added to show concordance; it is not a linear regression. (**a**) PcLiv = relative liver weight (percent). Rhombus: shows a statistically significant departure from Prop-RA for binary combination BDCM:CDBM (dosage = 3.0 mmol/kg/day, mixing ratio = 2.4:1). (**b**) LogALT = log10(ALT). Rhombus: shows a statistically significant departure from Prop-RA for binary combination CHCl_3_:CHBr_3_ (dosage = 3.0 mmol/kg/day, mixing ratio = 1:1). Square: shows a statistically significant departure from Prop-RA for binary combination CDBM:CHBr_3_ (dosage = 3.0 mmol/kg/day, mixing ratio = 10:1).

**Figure 5 toxics-12-00240-f005:**
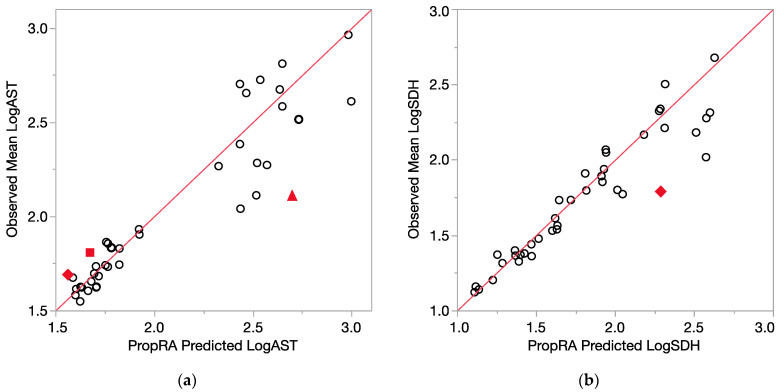
Consistency of data with the Prop-RA prediction for LogAST and LogSDH. Circles indicate concordance between data and Prop-RA prediction. Filled-in symbols denote statistically significant departure from the Prop-RA prediction for that dose–mixture combination. The red 45-degree line is added to show concordance; it is not a linear regression. (**a**) LogAST = log10(AST). Rhombus: shows a statistically significant departure from Prop-RA for binary combination CHCl_3_:CDBM (dosage = 0.1 mmol/kg/day, mixing ratio = 1:1). Square: shows statistically significant departure from Prop-RA for binary combination CHCl_3_:BDCM (dosage = 1.0 mmol/kg/day, mixing ratio = 2.7:1). Triangle: shows statistically significant departure from Prop-RA for binary combination CDBM:CHBr_3_ (dosage = 3.0 mmol/kg/day, mixing ratio = 10:1). (**b**) LogSDH = log10(SDH). Rhombus: shows a statistically significant departure from Prop-RA for binary combination BDCM:CHBr_3_ (dosage = 3.0 mmol/kg/day, mixing ratio = 1:1).

**Figure 6 toxics-12-00240-f006:**
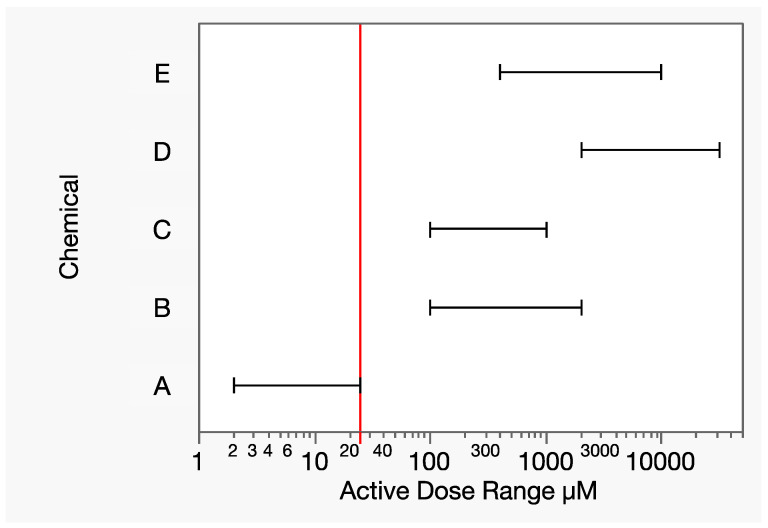
Dose ranges (horizontal line segments, excluding controls) of the hypothetical chemicals A–E for a given response of interest. The vertical line is the lowest maximum component dose level (25 µM), that of chemical A, at which 100% response was experimentally observed. There is no dose range overlap of chemical A with any other component, forcing an estimated response level based on possibly extreme extrapolation.

**Table 1 toxics-12-00240-t001:** Description of the binary experiments.

THMs ^1^	Molar Ratios (Fractions) ^2^	Total Dosage—Ratios (Individual THM Dosages), mmol/kg/day ^3^		Total Dosages, mmol/kg/day ^3^
CHCl_3_:BDCM	1:1 (0.5, 0.5) 2.7:1 (0.730, 0.270)	0.1–1:1 (0.05, 0.05)		Single THMs tested at 0, 0.1, 1.0, and 3.0 mmol/kg/day 1:1 mixture ratio tested at 0.1, 1.0, and 3.0 mmol/kg/day Environmentally relevant ratios tested at 1.0 and 3.0 mmol/kg/day
		1.0–1:1 (0.5, 0.5)	2.7:1 (0.73, 0.27)	
		3.0–1:1 (1.5, 1.5)	2.7:1 (2.19, 0.81)	
CHCl_3_:CHBr_3_	1:1 (0.5, 0.5) 65:1 (0.985, 0.015)	0.1–1:1 (0.05, 0.05)		
		1.0–1:1 (0.5, 0.5)	65:1 (0.985, 0.015)	
		3.0–1:1 (1.5, 1.5)	65:1 (2.955, 0.045)	
BDCM:CHBr_3_	1:1 (0.5, 0.5) 24:1 (0.960, 0.040)	0.1–1:1 (0.05, 0.05)		
		1.0–1:1 (0.5, 0.5)	24:1 (0.96, 0.04)	
		3.0–1:1 (1.5, 1.5)	24:1 (2.88, 0.12)	
BDCM:CDBM	1:1 (0.5, 0.5) 2.4:1 (0.706, 0.294)	0.1–1:1 (0.05, 0.05)		
		1.0–1:1 (0.5, 0.5)	2.4:1 (0.706, 0.294)	
		3.0–1:1 (1.5, 1.5)	2.4:1 (2.118, 0.882)	
CHCl_3_:CDBM	1:1 (0.5, 0.5) 6.5:1 (0.867, 0.133)	0.1–1:1 (0.05, 0.05)		
		1.0–1:1 (0.5, 0.5)	6.5:1 (0.867, 0.133)	
		3.0–1:1 (1.5, 1.5)	6.5:1 (2.601, 0.399)	
CDBM:CHBr_3_	1:1 (0.5, 0.5) 10:1 (0.909, 0.091)	0.1–1:1 (0.05, 0.05)		
		1.0–1:1 (0.5, 0.5)	10:1 (0.909, 0.091)	
		3.0–1:1 (1.5, 1.5)	10:1 (2.727, 0.273)	

^1^ There are replicate experiments for CHCl_3_:BDCM and BDCM:CDBM, which are referred to as CHCl_3_:BDCM-rep and BDCM:CDBM-rep, respectively. ^2^ Ratios use the same chemical ordering as column 1. A ratio a:b converts to component fractions a/(a + b) and b/(a + b). ^3^ Beginning dose group sizes = 12 for all experiments. Not all animals survived.

**Table 2 toxics-12-00240-t002:** Confidence intervals for relative liver weight (%) for binary combinations of BDCM:CDBM ^1,2^.

Dosage (mmol/ kg-day)	Ratio (BDCM: CDBM)	BDCM Obs	CDBM Obs	Mixture Obs	Mixture Pred	L (Linear Contrast)	Scheffé 95% Conf Interval
		Mean (SD) N	Mean (SD) N	Mean (SD) N	Mean		
0.1	1:1	4.69	5.24	5.05	4.96	−0.09	(−0.66, 0.48)
		(0.63)	(0.27)	(0.42)			
		7	7	7			
1.0	1:1	5.94	6.09	6.48	6.01	−0.47	(−1.42, 0.48)
		(0.44)	(0.48)	(0.24)			
		6	7	2			
3.0	1:1	8.78	5.86	8.05	7.32	−0.73	(−2.41, 0.96)
		(0.59)	(0.51)	(1.11)			
		5	2	3			
1.0	2.4:1	5.94	6.09	5.78	5.98	0.204	(−0.34, 0.74)
		(0.44)	(0.48)	(0.30)			
		6	7	7			
3.0	2.4:1	8.78	5.86	10.19	7.92	−2.27	(−4.11, −0.44) ^3^
		(0.59)	(0.51)	(1.33)			
		5	2	3			

^1^ Overall type I error rate for this study using Scheffé’s method is α = 0.05. ^2^ L = linear contrast; Obs = observed response; Pred = predicted response; confidence interval is for L; SD = standard deviation; and N = dose group size at study termination. ^3^ Result in the last row indicates a statistically significantly greater than Prop-RA response. It is the only instance of departure from Prop-RA out of the 40 comparisons from the eight binary experiments across all dosages and ratios for relative liver weight.

**Table 3 toxics-12-00240-t003:** All results for departures from Prop-RA confidence intervals for relative liver weight (%) and log-transformed serum enzyme data ^1,2^.

THMs (THM1: THM2)	Endpoint	Ratio	THM1	THM2	Mixture Obs	Mixture Pred	L (Linear Contrast)	Scheffé 95% Conf Interval ^3^
	(Dosage, mmol/kg/ day)		Mean (SD) N	Mean (SD) N	Mean (SD) N	Mean		
CHCl_3_: BDCM	AST (1.0)	2.7:1	1.64 (0.08) 9	1.78 (0.17) 7	1.81 (0.07) 11	1.68	−0.13	(−0.24, −0.02)
CHCl_3_: BDCM	For all endpoints, dosages, and ratios other than AST, 1.0 mmol/kg/day, 2.7:1 ratio, no departures from Prop-RA were detected.							
CHCl_3_: BDCM-rep	For all endpoints, dosages, and ratios, no departures from Prop-RA were detected.							
CHCl_3_: CHBr_3_	ALT (3.0)	1:1	2.71 (0.39) 8	2.40 (0.20) 8	2.10 (0.24) 4	2.56	0.46	(0.02, 0.90)
CHCl_3_: CHBr_3_	For all endpoints, dosages, and ratios, other than ALT, 3.0 mmol/kg/day, 1:1 ratio, no departures from Prop-RA were detected.							
BDCM: CHBr_3_	SDH (3.0)	1:1	2.53 (0.33) 5	2.05 (0.46) 4	1.79 (0.09) 6	2.29	0.50	(0.05, 0.95)
BDCM: CHBr_3_	For all endpoints, dosages, and ratios, other than SDH, 3.0 mmol/kg/day, 1:1 ratio, no departures from Prop-RA were detected.							
BDCM: CDBM	PcLiv (3.0)	2.4:1	8.78 (0.59) 5	5.86 (0.51) 2	10.19 (1.33) 3	7.92	−2.27	(−4.11, −0.44)
BDCM: CDBM	For all endpoints, dosages, and ratios, other than PcLiv, 3.0 mmol/kg/day, 2.4:1 ratio, no departures from Prop-RA were detected.							
BDCM: CDBM-rep	For all endpoints, dosages, and ratios, no departures from Prop-RA were detected.							
CHCl_3_: CDBM	AST (0.1)	1:1	1.57 (0.05) 8	1.56 (0.09) 8	1.69 (0.06) 7	1.56	−0.13	(−0.21, −0.04)
CHCl_3_: CDBM	For all endpoints, dosages, and ratios, other than AST, 1.0 mmol/kg/day, 1:1 ratio, no departures from Prop-RA were detected.							
CDBM: CHBr_3_	ALT (3.0)	10:1	2.82 (0.20) 5	2.19 (0.23) 4	2.19 (0.10) 2	2.76	0.57	(0.08, 1.06)
CDBM: CHBr_3_	AST (3.0)	10:1	2.74 (0.22) 5	2.30 (0.19) 4	2.11 (0.17) 2	2.70	0.59	(0.10, 1.08)
CDBM: CHBr_3_	For all endpoints, dosages, and ratios, other than ALT and AST, 3.0 mmol/kg/day, 10:1 ratio, no departures from Prop-RA were observed.							

^1^ Units of measurement are mmol/kg/day for dosage and IU/liter for the enzyme responses. PcLiv used untransformed dose units; serum enzyme data used log10-transformed data. ^2^ L = Linear contrast; Obs = observed response; Pred = predicted response; Conf Interval = confidence interval on L; SD = standard deviation; N = dose group size at study termination; and PcLiv = relative liver weight (%). ^3^ Overall type I error rate for this study using Scheffé’s method is α = 0.05.

**Table 4 toxics-12-00240-t004:** Comparison of original vs. log-transformed departure from Prop-RA analyses for the serum enzyme data, with homogeneity of variance (HOV) test results ^1^.

THMs	Endpoint (Dosage, mmol/kg/day)	THMs	Ratio	Original Analysis Scheffé CI ^2^ HOV Test *p*-Value ^3^	Log10 Analysis Scheffé CI ^2^ HOV Test *p*-Value ^3^
CHCl_3_: BDCM	AST (1.0)	CHCl_3_: BDCM	2.7:1	PRA ^3^ 0.002 *	(−0.24, −0.02) 0.003 *
CHCl_3_: CHBr_3_	ALT (3.0)	CHCl_3_: CHBr_3_	1:1	PRA 0.003 *	(0.02, 0.9) 0.04 *
BDCM: CHBr_3_	SDH (3.0)	BDCM: CHBr_3_	1:1	PRA 0.33	(0.05, 0.95) 0.35
BDCM: CDBM	ALT (3.0)	BDCM: CDBM	1:1	PRA 0.22	PRA 0.13
BDCM: CDBM	AST (3.0)	BDCM: CDBM	1:1	PRA 0.71	PRA 0.20
CHCl_3_: CDBM	AST (0.1)	CHCl_3_: CDBM	1:1	(−19.98, −4.76) 0.35	(−0.21, −0.04) 0.18
CHCl_3_: CDBM	SDH (0.1)	CHCl_3_: CDBM	1:1	(−13.19, −0.01) 0.12	PRA 0.10
CDBM: CHBr_3_	ALT (3.0)	CDBM: CHBr_3_	10:1	(2.98, 997.7) 0.28	(0.08, 1.06) 0.80
CDBM: CHBr_3_	AST (3.0)	CDBM: CHBr_3_	10:1	PRA 0.29	(0.1, 1.08) 0.82

^1^ The replicate experiments CHCl_3_:BDCM-rep and BDCM:CDBM-rep are not included in the table as no departures from Prop-RA were found for any of the above scenarios. Testing of departures from Prop-RA could not be made for BDCM:CDBM-rep because of the small survivor dose group size (N): N = 1 at dose = 3.0 for the mixture at the environmental ratio; and N = 1 at dose = 3.0 for CDBM and for the mixture at the 1:1 ratio. ^2^ The 95% confidence intervals are shown in parentheses. PRA means results were consistent with Prop-RA for that dosage/ratio/endpoint combination. If experimental results are not explicitly shown (i.e., excluded from this table), no significant departures from Prop-RA were found for those dosage/ratio/endpoint combinations. * = statistically significant (*p* < 0.05) for the O’Brien HOV test. ^3^ Value is the significance level for that dosage/ratio/endpoint combination.

**Table 5 toxics-12-00240-t005:** Comparison of predicted and observed means for mixtures significantly different from Prop-RA (enzyme means back-transformed from log10 response) ^1,2,3^.

THMs	Endpoint (Dosage mmol/kg/day)	Ratio	Mixture Obs Mean	Mixture Pred Mean	Mean Difference	Interaction Magnitude	Direction ^5^
					Pred-Obs	Pred to Obs ^4^	
CHCl_3_: BDCM	AST (1.0)	2.7:1	64.82	47.41	−17.41	1.4	>Prop-RA
CHCl_3_: CHBr_3_	ALT (3.0)	1:1	138.75	359.75	221.0	2.6	<Prop-RA
BDCM: CHBr_3_	SDH (3.0)	1:1	62.72	194.76	132.04	3.1	<Prop-RA
BDCM: CDBM	PcLiv (3.0)	2.4:1	10.19	7.92	−2.27	1.3	>Prop-RA
CHCl_3_: CDBM	AST (0.1)	1:1	49.43	36.34	−12.78	1.4	>Prop-RA
CDBM: CHBr_3_	ALT (3.0)	10:1	156.5	574.52	418.02	3.7	<Prop-RA
CDBM: CHBr_3_	AST (3.0)	10:1	133.5	501.56	368.06	3.8	<Prop-RA

^1^ For AST, ALT, and SDH, evaluation of departures from Prop-RA was conducted in log space. In this table, the means are back-transformed from log10 responses for ALT, AST, and SDH. ^2^ Units of measurement are mmol/kg/day for dosage, IU/liter for the enzymes, and % for relative liver weight. Obs = observed response; Pred = predicted response; and PcLiv = relative liver weight (%). ^3^ The 7 results in this table are the only instances of departure from Prop-RA out of 160 comparisons from the 8 binary experiments across all dosages and ratios for relative liver weight and the three serum enzyme endpoints. No departures from Prop-RA were found for CHCl_3_:BDCM-rep and BDCM:CDBM-rep. ^4^ Interaction magnitude is an n-fold change in response, i.e., max(Pred/Obs, Obs/Pred). ^5^ Direction indicates whether the deviation from Prop-RA was in the direction of greater-than predicted (>Prop-RA) or less-than predicted (<Prop-RA).

**Table 6 toxics-12-00240-t006:** Impact of daily dosage units on component fraction for the binary combination CHCl_3_:CHBr_3_ at a 1:1 mixing ratio so both component fractions are 0.5 ^1^.

Total Dosage mmol/kg	CHCl_3_ mg/kg (mmol/kg)	CHBr_3_ mg/kg (mmol/kg)	Total Dosage mg/kg	CHCl_3_ Converted Total Dosage ^2^ mmol/kg	Fraction CHCl_3_, mg-based ^3^
1.0	59.69 (0.5)	126.37 (0.5)	186.06	1.56	0.32
3.0	179.07 (1.5)	379.10 (1.5)	558.17	4.68	0.32

^1^ Component daily dose levels are converted to mg/kg from original mmol/kg values. The mmol/kg component dose levels are shown in parentheses in columns 2 and 3, calculated by (total dose level from column 1) × (component fraction of 0.5). The total dose level in mg/kg (column 4) is the sum of the converted component mg/kg dose levels. ^2^ Converted dose level for CHCl_3_ is the mixture total dose (mg/kg, column 4) converted to mmol/kg by dividing by the mol wt of CHCl_3_. ^3^ The mg-based fraction for CHCl_3_ is the mg/kg value in column 2 divided by the total dosage (mg//kg) in column 4.

## Data Availability

Data files for this project can be found at U.S. EPA’s ScienceHub website. Online. https://catalog.data.gov/dataset/epa-sciencehub (accessed on 5 March 2024).

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
