# Peer review of "Evaluation of a Proportional Response Addition Approach to Mixture Risk Assessment and Predictive Toxicology Using Data on Four Trihalomethanes from the U.S. EPA’s Multiple-Purpose Design Study"

_toxics, 2024, doi:10.3390/toxics12040240_

Round 1

Reviewer 1 Report

Comments and Suggestions for Authors

In the submitted manuscript the Authors have used the proportional response addition model to predict response from chemical mixture exposure, on the example of four trihalomethanes. The aim of the study was to validate and assess the chosen model. I found the manuscript interesting, well constructed and properly presented. However, I would also like the Authors to make several necessary corrections, listed below.

My main question is: what is the difference between the Prop-RA and well known weighted arithmetic mean?

I like the introduction, however, I would like to see another, very short paragraph, describing why the models, regardless of the choice, can sometimes fail or be inaccurate. This can result from the interactions between the mixture components, either the physicochemical interactions or at the pharmacokinetics (and dynamics too) level.

Lines 37-38, keywords should be divided by semicolons

Line 96, ATSDR abbreviation must be explained

Line 112, do you mean molar fraction?

Lines 131-132, it’s quite strange that for two compounds the molecular formulas are used and for two the abbreviations. Was it done purposely?

Lines 141-153, what about logP of those compounds? Is it similar?

The Authors should suggest the possible reasons behind such huge differences between the observed and calculated, using Prop-RA, values, shown in the figures 4 and 5, denoted as red points.

Reviewer 2 Report

Comments and Suggestions for Authors

The authors presented a model for predicting response from chemical mixture exposure. The Materials and Methods lack clarity and scientific precision. The experimental design must be written more clearly and accurately. Results should be strictly separate from the discussion. It is essential to integrate and interconnect all discussed tests in the discussion to lead to a general conclusion.

1. The abstract exceeds the limit of 200 words. It must be reduced.

2. In affiliations: complete address information including city, zip code, state/province, and country are missing. Tony McDonald is redundant to write in affiliation. I guess it is Anthony McDonald.

3. Is it necessary a dot after Eq. (1)?

4. line 159: At the first mention of the fraction (pj), as well as in line 203, it is not clear if is it the mass fraction, molar fraction, or volume fraction.

5. Line 208: It is not clear the meaning of yMIX = 0.34? What is the predicted mixture response? What is the unit?

6. Table 1. Ration: in which units? Is it mass, molar, or volume ratio?

7. Line 213? Authors have performed hepatotoxicity of the four THMs. What does mean series of sixteen experiments?

8. How many mice were used for each experiment? What does mean “experiments”? Experimental design must be written more clearly.

9. Line 244-245: How many replicated experiments were performed? There is no data about several surviving animals.

10. Section 2.4. Methods for Modeling the Data: If the derivations of the equations are not part of this research but are taken from the literature, a reference is sufficient.

11. Results: Authors presented the results of relative liver weight (PcLiv, as percent body weight) and the three log-transformed serum enzyme levels (ALT, AST and SDH) but the methods were not described in the Experimental part. How did animals were killed?

12. Authors must include details on housing, husbandry, and pain management in their manuscript. Authors should provide a statement justifying the work from an ethical perspective.

13. Line 454: The title of Table 3 must be completed.

14. Table 3: What does mean N (Dose Group Size) for THM1 and THM2? Is it several animals? Why does not equal?

15. Introduce the coefficient of correlation (R) into Figure 4 and Figure 5 and discuss.

16. Does Figure 6 present the results of this study or it is taken from the literature? If it is taken from the literature, it should be removed from the manuscript.

17. line 693: cols should be written colons

18. Table 6 should be transferred to the Results section.

Round 2

Reviewer 1 Report

Comments and Suggestions for Authors

The Authors have revised and improved their work. This version can be accepted.

Author Response

No comments from Reviewer 1 to comment on.

Reviewer 2 Report

Comments and Suggestions for Authors

The authors have improved the manuscript, but still minor errors were left:

·         Formula of bromoform (CHBr3) should be CHBr3 through the whole manuscript.

·         Formula of chloroform (CHCl3) should be CHCl3 through the whole manuscript.

·         There is no interpretation for the term 3 at the footnote of Table 4

·         Titles of the Figures 4 and 5 should be written in one paragraph. Legend is also not clear. Explain circles. Diamonds are actually rhombuses.

·         Symbols of the statistical parameters should be written italics.

Introduce space before and after = uniformly through the whole manuscript

Author Response

Reviewer 2  (Round 2)  comments

The authors have improved the manuscript, but still minor errors were left:
· Formula of bromoform (CHBr3) should be CHBr3 through the whole manuscript. - Done
· Formula of chloroform (CHCl3) should be CHCl3 through the whole manuscript. - Done
· There is no interpretation for the term 3 at the footnote of Table 4 - Reviewer is correct. We inserted the superscript 3 before the word "Value" in the legend.
· Titles of the Figures 4 and 5 should be written in one paragraph. Legend is also not clear. - Did not combine, but changed Figure  titles to include the endpoints shown in each Figure so that they are now different. Added more text to legends to clarify. 
Explain circles. Diamonds are actually rhombuses. - Changed text from Diamond to Rhombus.
· Symbols of the statistical parameters should be written italics. - Done.
Introduce space before and after = uniformly through the whole manuscript -  Done.

Round 3

Reviewer 2 Report

Comments and Suggestions for Authors

The paper is ready for publication.